# Underground Ink: Printed Electronics Enabling Electrochemical Sensing in Soil

**DOI:** 10.3390/mi15050625

**Published:** 2024-05-07

**Authors:** Kuan-Yu Chen, Jeneel Kachhadiya, Sharar Muhtasim, Shuohao Cai, Jingyi Huang, Joseph Andrews

**Affiliations:** 1Department of Electrical and Computer Engineering, University of Wisconsin-Madison, Madison, WI 53706, USA; kchen372@wisc.edu (K.-Y.C.); jeneel.kachhadiya@wisc.edu (J.K.); smuhtasim@wisc.edu (S.M.); 2Department of Soil Science, University of Wisconsin-Madison, Madison, WI 53706, USA; scai62@wisc.edu (S.C.); jhuang426@wisc.edu (J.H.); 3Department of Mechanical Engineering, University of Wisconsin-Madison, Madison, WI 53706, USA

**Keywords:** printed electronics, electrochemical sensors, soil monitoring, data integration

## Abstract

Improving agricultural production relies on the decisions and actions of farmers and land managers, highlighting the importance of efficient soil monitoring techniques for better resource management and reduced environmental impacts. Despite considerable advancements in soil sensors, their traditional bulky counterparts cause difficulty in widespread adoption and large-scale deployment. Printed electronics emerge as a promising technology, offering flexibility in device design, cost-effectiveness for mass production, and a compact footprint suitable for versatile deployment platforms. This review overviews how printed sensors are used in monitoring soil parameters through electrochemical sensing mechanisms, enabling direct measurement of nutrients, moisture content, pH value, and others. Notably, printed sensors address scalability and cost concerns in fabrication, making them suitable for deployment across large crop fields. Additionally, seamlessly integrating printed sensors with printed antenna units or traditional integrated circuits can facilitate comprehensive functionality for real-time data collection and communication. This real-time information empowers informed decision-making, optimizes resource management, and enhances crop yield. This review aims to provide a comprehensive overview of recent work related to printed electrochemical soil sensors, ultimately providing insight into future research directions that can enable widespread adoption of precision agriculture technologies.

## 1. Introduction

The success of agricultural production hinges on the decisions and practices of farmers and land managers worldwide. This is true across scales, from small-scale subsistence farming to large industrial operations, with both navigating challenges posed by nature and market fluctuations [1]. Monitoring soil conditions (e.g., moisture, nutrients, and pollutants) over growing seasons enhances resource efficiency, ultimately leading to maximized agricultural yields while simultaneously minimizing environmental impacts, as shown in Figure 1. Furthermore, long-term field observations of soil conditions along with crop yield, weather records, and management practices contribute to the databases that elucidate the intricate relationships between plant growth and environmental conditions. Data-enabled insights can further improve the mechanistic modeling of crop responses to climate change [2], breeding for stress-tolerant varieties [3], and the advancement of intelligent and sustainable automated agricultural systems [4].

Soil sensors are critical components that enable applications in smart and precision agriculture [5]. Novel soil sensing technologies are dedicated to satisfying various attributes. In order to promote wide adoption of soil sensors in field practices, it is necessary to demonstrate high sensing accuracy and reliability for crucial soil parameters (e.g., water content, nutrient levels, gas composition, heavy metal concentrations) with sufficient spatiotemporal resolution. In this era of data-driven agriculture, Sensors 4.0 technologies enable sensor networks and advanced algorithms for data processing in smart farming [6,7]. The development of low-power wireless sensor networks with robust data processing and long-range communication capabilities is essential [8]. Versatile soil sensing platforms must be created for large-scale deployment to continuously collect real-time soil microenvironment data [9]. The rising need for soil sensing devices drives the development of cost-effective, reliable, and maintenance-free self-powered/power-independent sensors and integrated platforms [10].

Printed electronics can be a key enabling technology for developing multiple modalities of soil sensors. Specifically, printed electronics can be defined as a fabrication technique that allows for spatial patterning of a wide variety of liquid electronic inks, which can include conductors, semiconductors, and insulators [11]. The technique is highly beneficial when pursuing electronics that are large-area [12,13,14], flexible [15,16], and low-cost [17]. These attributes specifically benefit soil sensing applications. For example, sensing a relatively large but representative area is required due to the heterogeneity of the soil. Additionally, flexible form factors can be capitalized on for conformally applying planarly fabricated devices to non-planar surfaces, such as cylindrical rods. Lastly, the low-cost fabrication technique is highly scalable, allowing for cost-effective production. As depicted in Figure 2, printing techniques have been demonstrated for a wide array of electrochemical sensors for soil sensing applications. These include voltametric, amperometric, potentiometric, and impedimetric sensors. In addition to printing active sensor elements, printed devices can be designed to be incorporated with traditional integrated circuits (ICs), generating flexible hybrid electronic circuitry with complete functionality [18]. Overall, these hybrid systems can offer sensing capabilities combined with data aggregation and communication functions, allowing for a fully functional sensing system. These advantages make printed electronics a promising candidate for applications In agriculture.

In this review, we present a summary of recent advancements in printed electrochemical soil sensors, including their implementation of hybrid sensor network systems. Additionally, we describe the potential applications that are enabled by printed electronics to address future agricultural needs. This review aims to highlight the role of printed electronics in soil monitoring, with a specific focus on printed electrochemical sensors, and to address the current gap in real-time soil measurement within agricultural applications. For readers interested in exploring printed sensors for agricultural applications beyond the scope of electrochemical sensing mechanisms, we suggest referring to the review titled “Printed Sensor Technologies for Monitoring Applications in Smart Farming: A Review” [19].

## 2. Printed Electrochemical Sensors for Soil Sensing

Electrochemical sensors are employed in soil for real-time monitoring of crucial soil parameters [20]. Electrochemical sensors can be categorized by their detection mechanisms, including potentiometric (measuring the electrical potential difference between two electrodes), voltametric (which measures the resulting current from varying the applied voltage), amperometric (measuring the current generated by oxidation or reduction reactions at electrode surfaces), and impedimetric (assessing the impedance to alternating current) sensors [21]. This diversity of sensor types allows for monitoring of pH levels, nutrient concentrations, moisture content, and pollutant concentrations, as illustrated in Figure 2.

Electrochemical soil sensors can be fabricated using both contact and contactless printing techniques [19]. Screen-printing technology is a mature and widely adopted contact printing method in sensor fabrication and allows for large-area high throughput patterning [22,23,24]. Contactless printing methods, such as inkjet printing and aerosol jet printing, offer distinct advantages for printing on a wide range of substrates. Inkjet-printed sensors [25,26,27] and aerosol jet printing sensors [28] enable digital customization of patterns and offer high trace resolution. Additionally, other printing techniques, such as flexographic printing or gravure printing, can also be utilized to fabricate electrochemical sensors, each offering unique benefits [19]. Consequently, the choice of printing technique depends on capabilities in terms of production throughput, pattern resolution, and desired sensor characteristics. Advanced printing techniques enable a cost-effective fabrication process for electrochemical sensors, thereby enhancing accessibility for widespread deployment in agricultural land and environmental monitoring systems. Furthermore, the simplicity and non-bulky nature of printed sensors allow integration into proximal tools and handheld devices for on-site soil analysis. This section provides an overview of the work on printed electrochemical sensors for the direct analysis of nutrients, pollutants, moisture content, and other physicochemical parameters in soils.

### 2.1. Potentiometric Sensors

The potentiometric sensor is one of the most studied electrochemical sensors that can transduce chemical activity into a measurable voltage signal [29,30]. Generally, potentiometric sensors are composed of two electrodes: a working electrode coated by an ion-selective membrane, where only target ions can permeate through the membrane, and an inner reference electrode submerged in a liquid electrolyte [31]. The potential difference between the electrodes represents the target ion concentrations and can be described using the classical Nernst equation [31,32,33]:(1)$$E=E_{0}+\frac{\mathrm{RT}}{\mathrm{nF}}\mathrm{lnQ}$$
where E is the cell potential (V); E_0_ is the cell potential under standard conditions (V), R is the universal gas constant, 8.314 J/(mol·K), T is the temperature in K, n is the number of electrons transferred in the reaction, F is the Faraday constant, 96,485 C/mol, and Q is the reaction quotient. According to the Nernst equation, the sensitivity of an ideal ISE for the change of an order magnitude in the concentration of a monovalent ion (n equals 1) at ambient temperature (25 °C) is around −59.1 mV/dec [34,35].

Conventional potentiometric sensors require frequent maintenance and calibration due to their inner filling solutions, restricting their use to laboratory settings [29]. Solid-state ion selective electrodes (ISE) eliminate the need for inner filling solutions, thus simplifying sensor design [30]. Additionally, they fulfill the demand for portable devices by requiring small sample volumes, being easy to use, and providing stable potential readings. Table 1 shows the utilization of potentiometric sensors in environmental analysis applications, particularly in soil monitoring. Sensitivity, linear detection range, and limit of detection (LOD) serve as key figures of merit. Sensitivity determines the resolution of detection, while linear detection range and LOD indicate the sensing range and lowest concentration above background noise, respectively. A key application for potentiometric sensors is in soil testing, as farmers and/or land managers can utilize spatially resolved soil nutrient measurements to inform nutrient needs and site-specific fertilization protocols [36]. Another key task in soil testing is determining the real-time pollution levels of some chemicals, especially heavy metals, in the soil, which is important for investigating the effects of soil pollution on both the environment and human/livestock health [37].

Lemos et al. developed a potassium ISE potentiometric sensor for agricultural purposes. The findings show that there is a good linear relationship between probe measurements both in the field and in-lab analysis and present sensitivity within the range of 69 to 71 mV/dec [42]. However, their sensor relies on standard potassium analytical solutions inserted in the probe to do the calibration, leading to increased sensor costs. Cranny et al. developed a system for chloride measurement in soils, where potentiometric sensors are inserted in a soil column and tension infiltrometers are used on the top of the soil to control the flow rates of chloride solutions [35]. This sensor system responds to chloride ions over a range of 3.55 ppm to 7091 ppm with a sensitivity of −49.8 ± 1.7 mV/dec.

The emergence of printed potentiometric sensors benefits from additive manufacturing, enabling the fabrication of cost-effective, miniaturized sensor systems [43,44]. Ali et al. introduced a nanocomposite of poly(3-octyl-thiophene) and molybdenum disulfide (POT-MoS_2_) into a working electrode in order to increase electron conductivity and anion exchange rate [39]. The sensors were embedded in soil slurry with a sensitivity of 64 mv/dec for nitrate-nitrogen detection. Chen et al. developed a nitrate potentiometric sensor with a porosity polyvinylidene fluoride (PVDF) cover, as depicted in Figure 3 [26]. This innovation allows for direct nitrate measurements in sandy soil and silt loam soil with a sensitivity of 56.3 mV/dec and 46.6 mV/dec, respectively. Additionally, the study successfully measured nitrate at varied water content levels in soil, giving a sub-Nernstian sensitivity of 39.1 mV/dec in 10% *w*/*w* sandy soil and 27.1 mV/dec in 25% *w*/*w* silt lam soil.

Reproducibility limits the use of solid-state ISE sensors in field applications due to the necessity of calibration [34]. Recent research has demonstrated the viability of solid-state ISE for extended periods, ranging from a month [39] to over a year [35]. Ali et al. coat the reference electrode with a Nafion membrane, resulting in a constant potential reading for 32 days. The Nafion membrane blocks negative ions while preventing Cl^−^ leaking from the Ag/AgCl electrode. Moreover, unlike controlled laboratory conditions, the sensitivity of sensors in field environments can be greatly affected by variability in soil temperature, moisture content, and soil textures [34,41,45]. Zhu et al. propose an evaluation method for the potentiometric sensors, assessing sensors made by Ali et al. [39], through various tests conducted under different soil temperatures and water content [45]. Furthermore, interference from other ions present in soils poses another significant issue for ISE sensors [36]. Selectivity studies carried out by Baumbauer et al., reveal that high concentrations of calcium can influence nitrate sensor potential readings [41]. Consequently, additional calibration standards are required for calcium-rich soils. Future research directions should prioritize enhancing the reproducibility of sensors, simplifying the calibration process, and exploring novel materials for ion-selective membranes. These efforts will contribute to the advancement of solid-state ISE sensors in field settings.

### 2.2. Voltametric Sensors

The voltametric method allows for advantages including high sensitivity, rapid detection speed, high accuracy, and cost-effectiveness. These advantages are key for detecting ions in soil [46]. Various voltammetry techniques have been used for detecting ions in soil, such as cyclic voltammetry (CV) [23,24,47,48,49,50,51,52,53,54,55,56], square wave voltammetry (SWV) [50,57,58,59], and differential pulse voltammetry (DPV) [60]. While all of these techniques involve sweeping a voltage across the working electrode as a function of time, the waveform differs for each voltametric technique. CV is popular for its simplicity in measurements and effectiveness in detecting nutrients in soil, making it the most popular method; DPV and SWV, however, offer higher sensitivity for ion detection [61].

#### 2.2.1. Cyclic Voltammetry

CV has been widely used in soil sensing research due to its ease of interpretation in investigating the oxidation and reduction of different molecular species, as well as studying transfer-initiated chemical reactions [62]. The potential of a working electrode is scanned linearly in both forward and reverse directions while measuring the corresponding current. As the potential is scanned in the forward direction, a solution containing the reduced form of a redox couple undergoes oxidation. At the switching potential, the direction of potential scanning is reversed, resulting in the reduction of the oxidized solution back to its reduced form. The peak current observed in CV is a result of analyte species diffusion, which happens when the analyte is depleted on the electrode surface [63]. The peak current changes proportionally with the concentration of the analyte, as described by the Randles-Ševčík equation [62]:(2)$$I_{p}=2.69\;\times \;10^{5}n^{\frac{3}{2}}{\mathrm{AD}}^{\frac{1}{2}}{\mathrm{Cv}}^{\frac{1}{2}}$$
where I_p_ is the peak current, n is the number of electrons transferred during the redox reaction, A is the surface area of the electrode in cm^2^, C is the concentration in mol m L^−1^, D is the diffusion coefficient of the species in cm^2^s^−1^, and v is the scan rate in Vs^−1^. The proportional relation allows for quantification of analytes without a complicated calibration process, making CV a simple and efficient ion sensing method. Recently, CV has been used in numerous applications for detecting nutrients in soil using printed sensor techniques, as shown in Figure 4.

Chen et al. developed cadmium sulfide nanorods modified screen-printed electrodes (CdS NRs-SPE), coupled with a portable potentiostat and an extraction filter that could detect nitrate levels in soil samples in less than 10 min [23]. The reference electrode was fabricated using silver/silver chloride (Ag/AgCl), and the working and counter electrodes were fabricated using carbon paste, onto which cadmium sulfide nanorods were drop-casted. The limit of detection (LOD) of the sensor was 2.3 µM with a linear range of 0.05–5 mM in real soil samples, and the detection results were within 96% of measurements carried out using standard methods. Nitrate detection using cyclic voltammetry has also been carried out by Kundu et al. by using commercial screen-printed sensors onto which multiwalled carbon nanotubes (CNT) and hematite nanoflowers (α-Fe_2_O_3_) were drop-casted [47]. Surface modification of the electrodes was performed using the nitrate reductase (NiR/CNT-α-Fe_2_O_3_/SPE) enzyme and bovine serum albumin (BSA/NiR/CNT-α-Fe_2_O_3_/SPE). The latter displayed higher reproducibility and precision in clay loam, sandy loam, and silt loam soil samples with less than 2% relative standard deviation compared to colorimetric methods. Yin et al. developed and constructed a dual-printed sensor for detecting nitrate and moisture using silver nanoparticle inks on Kapton films with aerosol jet printing. The nitrate sensor was able to detect nitrate levels from 1 to 400 mg/mL using cyclic voltametric techniques with a scan rate of 150 V/s [48].

CV can also be effective in the detection of nitrites. Although different materials such as glassy carbon, gold, platinum, and metal oxides have been implemented in the detection of nitrates, they are less practical in real-world conditions due to the fouling of sensor surfaces with different species [64,65]. The use of nanomaterial technology in electrochemical sensors can help improve the sensitivity and selectivity of nitrite by enhancing electroanalytic activity and improving electron kinetics [66]. Gurban et al. designed a nitrite sensor based on screen-printed carbon paste electrodes modified with multiwalled carbon nanotubes and chitosan (MWCNT-CS/SPE) [49]. CV was used to test the effect of pH and MWCNT loading in the composite material before amperometric tests were carried out in nitrite solutions and in soil. The sensor obtained recovery values between 107.7% and 112.9%. Pal et al. constructed a 3D-printed nitrite sensor consisting of a non-conductive polylactic acid (PLA) cell and conductive carbon-loaded PLA filament electrodes. This sensor detected nitrites in both water and soil samples from 25 to 75 µM with an LOD of 1.96 µM [50].

Phosphorus is another key element in providing nutrition to plants in soil and water, but excessive levels of phosphorus can cause eutrophication. Therefore, monitoring real-time phosphorus levels is important in guiding phosphorus application and informing water quality. ZrO_2_ and ZnO are metal oxides with electrical and chemical properties that allow phosphate analytes to penetrate electrodes more efficiently due to their high affinity with phosphates. Lu et al. combined this principle with the use of multiple wall carbon nanotubes (MWCNTs) and ammonium molybdate tetrahydrates (AMT) to screen-print ZrO_2_/ZnO/MWCNT/AMT/SPE electrodes that generate electroactive species between the Mo(VI) and phosphate anions [51]. These species were tested using cycling voltammetry to determine trace amounts of phosphate and yield a limit of detection of 2 × 10^−8^ mol L^−1^. However, these electrochemical phosphate sensors are limited to single use and cannot be used for continuous monitoring of soil samples. Tang et al. devised an on-site continuous phosphate monitoring system in soil consisting of screen-printed electrodes modified with carbon black nanoparticles (CBNPs) [52]. The electrodes were embedded with polydimethylsiloxane (PDMS) channel layers and substrates that reduced the leakage between the electrodes and flow cell. CV was used to test the sensor at phosphate concentrations ranging from 5 µM to 400 µM by forming phosphomolybdate complexes, but the current response declined for phosphate concentrations greater than 100 µM. The sensor was used for continuous monitoring of soil for 2 h in field conditions 30 times. CV has also shown promise for sensors using unconventional substrates as well. For instance, Cioffi et al. developed a sensor with wax-printed and screen-printed electrodes on common office paper, which were dropcasted with Prussian blue, carbon black, and butyrylcholinesterase for detecting organophosphorus pesticides in agricultural soil [53]. Although the performance of the sensor was evaluated using chronoamperometric methods, Cioffi et al. used cyclic voltammetry to optimize the parameters of the sensors and test the substrate’s electrochemical effectiveness.

Potassium is also an essential nutrient for plants in agriculture; lower levels of potassium inhibit growth and reduce yield, whereas higher levels of potassium affect seed germination and prevent the uptake of other minerals, which makes monitoring the levels of potassium important. Bhandari et al. used screen-printed carbon electrodes modified with Nafion and functionalized with 4-aminobenzo-18-crown-6 ether to detect the presence of potassium ions in soil [54]. Both cyclic voltammetry and differential pulse voltammetry were used to test the performance of the sensor in soil. The presence of 18-crown-6 ether allowed for host-guest recognition of K^+^ ions over Na^+^, Ca^2+^, and NH_4_^+^ ions, allowing the sensing interface to detect potassium ions in the range of 1 to 500 ppm. Potassium ions have also been detected in conjunction with other ions and parameters, making them more suitable for on-field applications. A multisensor array was proposed by Sophocleous et al. that could measure potassium concentrations in soil with a sensitivity of 0.6 µA/mM, along with other parameters such as nitrate levels, pH, and temperature [24].

Printed voltametric sensors have also been demonstrated in the detection of heavy metal ions, such aslead (Pb), cadmium (Cd), copper (Cu), and mercury (Hg). Zhang et al. used screen-printed electrodes modified with gold (Au) nanoparticles and polypyrrole (Au@Py), onto which the complementary strand of the aptamer of Pb^2+^ (was adsorbed and then combined with the aptamer [55]. This biosensor was able to detect Pb^2+^ ions in the range of 0.5–25 ppb. Kadara et al. reported an electrochemical sensor consisting of electrodes prepared by mixing 2% bismuth oxide (Bi_2_O_3_) with graphite-carbon ink [56]. Electrochemical measurements were carried out in soil extracts and wastewater samples for the detection of Pb(II) and Cd(II) using cyclic voltammetry and chronopotentiometry and showed detection limits of 8 and 16 µg L^−1^, respectively. However, this sensor was unable to simultaneously detect both lead and cadmium. Detecting multiple ions simultaneously using CV can be challenging due to peak overlaps of different metal ions with similar redox potentials [61]. Anodic Stripping Voltammetry (ASV) is often used for simultaneous detection due to its capability to identify multiple ions according to each formal redox potential, as further discussed in Section 2.2.2 [67].

#### 2.2.2. Anodic Stripping Voltammetry

ASV is another popular sensing technique in voltammetry, particularly for the detection of heavy metal ions [61,68,69,70,71]. In ASV, metal ions undergo reduction to their zero-valence state on the electrode through electrodeposition by applying negative potentials—a process known as preconcentration. Next, the metal undergoes oxidation upon application of a positive potential, releasing the metal ions back into the solution and generating a current [68]. Faucher et al. developed a method to detect copper (Cu) in agricultural soils that are contaminated by lead by modifying carbon screen-printed electrodes with mercury deposition, which facilitates the preconcentration step [57]. Cinti et al. modified screen-printed electrodes by drop-casting with carbon black-gold nanoparticles (CBNP-AuNP) to detect Hg^2+^ ions. This modification takes advantage of the high affinity of AuNP for mercury as well as the high surface area of CBNP for better dispersion [58]. The sensor was tested in river water as well as in soil samples, but only a 56% recovery rate was obtained. Using electrochemically assisted self-assembly (EASA), Lv et al. prepared modified screen-printed carbon electrodes with silica isoporous membranes (SIM), whose high surface area allows for enhanced SWASV signals and has anti-fouling capabilities due to steric exclusion and selective permeation [59]. The sensor was used for the simultaneous detection of Cd^2+^, Pb^2+^, Cu^2+^, and Hg^2+^ ions [59]. Wang et al. utilized screen-printed electrodes modified with Nafion polymer and bismuth film with differential pulse ASV to detect trace amounts of Cd^2+^ and Pb^2+^ ions in soil, achieving limits of detection of 1.6 and 2.5 µg L^−1^, respectively [60].

Table 2 summarizes printable voltametric sensors used for soil applications in measuring nutrients and detecting heavy metals based on the ion type, range, limit of detection, and sensitivity. Voltametric sensors showed linearity relations, low detection limits, high sensitivity, and a short response in soil sensing applications. Furthermore, the adaptability of voltammetry to a variety of electrode materials enhances its capability to detect various types of ions. This versatility makes printed voltametric sensors a promising tool in precision agriculture and environmental sensing.

### 2.3. Amperometric Sensors

Amperometry is a technique used to measure the faradaic current produced by a constant applied to the reducing or oxidizing potential. In this electrochemical process, the oxidation or reduction of an electro-active species involves the transfer of electrons, leading to a diffusion-controlled current [72]. The process can be expressed by the following equation:(3)$$I=\frac{{\mathrm{nFAD}(C}_{\mathrm{bulk}}-C_{x=0})}{\delta }$$
where I is the controlled current, n is the electron transfer number, A (cm^2^) is the surface area of the electrode, D (cm^2^/s) is the analyte diffusion coefficient diffusion, F (C/mol) is the faraday constant, and $C_{\mathrm{bulk}}$ and $C_{x=0}$ (mol/cm^3^) is the concentration of the analyte in the bulk solution and at the surface of the electrode, respectively, and δ (cm) is the thickness of the diffusion layer [73]. Amperometry emerges as a practical solution for on-side analysis due to its simplistic instrumentation, minimal sample preparation requisites, and quick response time make amperometry a feasible choice for on-site analysis. The broad sensing range and selective detection capabilities of amperometry across various analytes are attributed to the precise control of applied potential and electrode modification.

Enzyme inhibitor amperometry utilizes enzymes as recognition elements within electrochemical sensors for targeted analyte detection [74,75]. Enzymes facilitate reactions that yield detectable alterations in current or voltage. Once the enzyme’s function is impeded by the presence of the analyte, there is a reduction in the measured current or voltage, as shown in Figure 5a. By monitoring this decrease, the concentration of the analyte can be determined. Gurerrieri et al. employed the drop-casting method to immobilize acetylcholinesterase (AChE) onto a screen-printed platinum electrode with the aim of detecting ethyl parathion from soil extracts [76]. Organophosphate esters inactivate AChE on the screen-printed sensor, resulting in a reduced current reading. In a separate study, Cioffi et al. determined the pesticide by measuring the electrodeactive byproduct (thiocholine) of butyrylcholinesterase (BChE) [53]. Moreover, a nanohybrid composite of BChE mixed with carbon black and Prussian Blue was applied to the working electrode to increase conductivity.

To enhance the stability of enzyme inhibitor biosensors, Sok and Fragosa modified the working electrode by conjugating tyrosinase to carbon nano-onions (CNOs) embedded in a chitosan matrix for glyphosate detection [77]. Glyphosate, a widely used herbicide, is classified as an organophosphate. Figure 5d shows that addition of CNO nanomaterials serves to increase the active surface area, promote electron transfer through conductive nanoparticles, and improve sensor stability [77]. Modifying the working electrode of an amperometric sensor with nanomaterials is a popular method for enhancing performance [72,75,77]. This modification enhances enzyme interaction by increasing surface area, while the high electrical conductivity of nanomaterials facilitates electron transfer, resulting in better sensing capabilities.

Amperometry has certain limitations for continuous measurements. Sensitivity gradually decreases over time due to the accumulation of reaction products on the electrode surface. Ensuring enzyme activity over an extended period poses another difficult challenge. Nevertheless, the portability of screen-printed electrodes makes amperometry devices suitable for hand-held sensors and proximal applications [78]. The screen-printed devices are cost-effective and disposable after use [53,76,77]. Consequently, farmers can efficiently utilize these devices for the periodic determination of heavy metals and pesticides.

### 2.4. Impedimetric Sensors

Among electrochemical analysis methods, the impedimetric technique has been significantly explored. Impedimetric analysis relies on alternating-current (AC) analysis, utilizing single, multiple, or sweeping frequency scanning techniques [79]. Electrochemical Impedance Spectroscopy (EIS) sensors can analyze the analyte by measuring the ratio of voltage to current at varying frequencies [80]. Conventional EIS requires an LCR meter for non-faradic assays. However, LCR meters are bulky and therefore not suitable for portable applications or handheld devices for real-time field measurements. Printed EIS sensors can be easily interfaced, however, with integrated electronic units, enabling the assembly of a compact measurement platform that includes a microcontroller, SD card, battery, and potentiostat. Eldeeb et al. utilized a screen-printed planar three-electrode sensor platform to conduct in situ EIS tests in soil, as shown in Figure 6 [81,82]. Customized films can be applied to the working electrode for different analytes. Eldeeb et al. drop-casted a nitrate ion-selective film on the working electrode, enabling the detection of nitrate in sandy loam, clay, and loamy clay soil in the range of 6−64 ppm with a decrease of approximately 900 Ω, 650 Ω, and 900 Ω per 5 ppm increase, respectively [81]. The working electrode can also be coated with a mixture of alizarin and Nafion to detect the soil pH value, demonstrating a sensitivity of 258.5 Ω/pH in clay soil [82].

Another popular device architecture for EIS measurements includes an interdigitated electrode (IDE) pattern. IDE sensors have two closely spaced interdigitated electrodes, consisting of alternating fingers. The planar geometry of IDE makes them suitable for fabrication using printing techniques. Aliyana et al. developed a screen-printed disposable pH sensor on a paper substrate. The IDE was fabricated using graphene-carbon ink. The biocompatible sensor allows for real-time soil measurement in the pH range 2−8 with a high sensitivity of 5.27 kΩ/pH. Korek et al. fabricated an impedimetric potassium sensor with all the sensor components made by inkjet printing, including IDE and ion selective membrane (ISM) [83]. The sensors show a sensitivity of 4.553 KMΩ/(mmol/L) for potassium ions in soil.

Conductometric sensors are a particular subset of impedimetric sensors [84]. The conductometric technique employs an AC current to detect the capacitance value. Figure 7 shows that many studies have demonstrated the use of IDE sensors for soil humidity measurements by maximizing the capacitance per unit area, resulting in greater sensitivity to changes in humidity levels [85,86,87]. Sui et al. make a biodegradable soil moisture sensor that allows real-time soil moisture measurements [88]. Biswas et al. found that the sensitivity of IDE soil moisture sensors is correlated with the width of electrodes both experimentally and through simulation [89].

### 2.5. Other Sensors

In previous sections, we have observed many research works showing printed soil sensors capable of detecting soil nutrient information using various electrochemical methods, including potentiometric, voltametric, amperometric, and impedimetric approaches. Apart from these established electrochemical sensing methods, printed electronics offer the potential to sense soil information through unconventional technologies. An alternative method involves utilizing colorimetry to differentiate analyte concentrations based on light intensity [27,90]. Thongkam et al. developed a screen-printed paper-based analytical device (PAD) for detecting ammonium in soil by employing Berthelot’s reaction on the PAD [91]. The color intensity is linearly related to the ammonium concentration in soil. Thongkam and Hemvibool further modified the PAD into a 3D PAD for detecting phosphate in the soil using the molybdenum blue method in a colorimetric assay [92].

Soil strength is inherently influenced by soil properties such as texture, structure, and bulk density. Measuring soil pressure aids in understanding the stress during tillage, seeding, and irrigation. Hong et al. employed the Frequency Domain Reflectometry (OFDR) method to detect soil pressure using 3D-printed optic pressure sensors made from polylactic acid filament [93]. In another study, Sophocleous et al. discovered that the correlation between soil electrical conductivity and water content can be utilized to identify soil structure [94].

Soil temperature is a key variable that affects agricultural growth and productivity [28]. Temperature sensors are employed in the soil for accurate monitoring of temperature. These sensors typically utilize a probe-based approach to acquire temperature readings, which are later converted from analog to digital signals for data processing and analysis. Sui et al. developed a simple inkjet-printed silver-based thermistor, giving a temperature sensitivity of 0.25 Ω/°C [95]. Furthermore, the straightforward structure of this resistance-based thermistor shows potential for integration with wireless accessories for communication. By utilizing wireless communication platforms, such as IoT (Internet of Things) networks or cloud-based systems, real-time soil temperature data can be seamlessly transmitted and accessed remotely [96].

## 3. Peripheral Electronics for Measurement and Monitoring

Peripheral electronics are pivotal to data acquisition in soil sensing, serving as the link between the sensor and the data processing unit. These electronics facilitate the conversion of analog signals from the sensor into digital data, enabling subsequent processing and analysis. This conversion process is crucial for users to efficiently achieve accurate primary soil information, laying the groundwork for precision agriculture.

### 3.1. Data Communication and Transmission

Electrochemical sensors typically output analog signals, such asvoltage or current, which represent the measured parameter. In this section, we explore the various aspects of connecting with these sensors, including the transmission of acquired data and data storage for further analysis.

Converting the analog output signal into a proper format is necessary for processing and transmission. This often requires signal conditioning circuits to amplify, filter, and digitize the sensor output. Traditional data acquisition systems used for the characterization of sensors typically involve wired connections or manual readings using LCR meters or similar equipment [48,97]. For a more seamless solution, microcontrollers, such as Arduino [81,98,99], ESP32 [100], or Raspberry Pi [101], can be directly incorporated into sensor designs to process and transmit data. Printed electronics offer flexible and customizable solutions for integrating signal conditioning components directly onto sensor substrates, which reduces the need for external circuitry and enhances overall system integration.

Storing the acquired sensor data is essential for long-term analysis, trend identification, and decision-making. Printed electronics enable the integration of data logging functionalities directly onto sensor nodes, allowing for onboard storage of sensor readings. Sensor designs can incorporate flash memory or microSD card slots for local data storage. Furthermore, wireless data transmission to cloud-based storage platforms enables centralized data management and access from any location with an internet connection. Cloud storage solutions offer scalability, redundancy, and accessibility, making them ideal for large-scale soil monitoring networks [102].

### 3.2. Power Management for Soil Sensors

In the realm of soil sensing, ensuring consistent and reliable power for sensor nodes is paramount for uninterrupted data acquisition and transmission. This section delves into power management strategies used in printed electronic sensor systems made for soil monitoring applications.

Traditional power sources such as batteries pose challenges in terms of limited lifespan, labor-intensive maintenance requirements, and adverse environmental impacts. To address these concerns, energy harvesting technologies offer promising solutions for powering soil sensors sustainably. Energy harvesting mechanisms, such as solar cells [24,103,104], and thermoelectric generators [105,106], leverage ambient energy sources in the soil environment to generate electrical power.

Efficient power management circuits are essential for optimizing energy utilization and prolonging the operational lifespan of soil sensor nodes. These circuits regulate voltage levels, manage energy storage, and control power distribution to different components [107]. Power management strategies, such as duty cycling, sleep modes, and voltage regulation, minimize power consumption during idle periods, thereby extending battery life or reducing the dependence on external power sources.

Radio Frequency Identification (RFID) technology presents a compelling solution for batteryless sensor systems in soil monitoring applications. RFID tags, powered by electromagnetic fields emitted by RFID readers, eliminate the need for onboard power sources, making them ideal for low-power, long-term deployments in remote environments. RFID-enabled sensor nodes can be embedded into the soil and place their RFID readers above ground for wireless communication [108]. This setup enables real-time data acquisition without the limitations imposed by traditional wired connections or battery-powered sensors. Leveraging RFID technology, battery-less sensor systems offer scalable and cost-effective solutions for continuous soil monitoring, thus facilitating data-driven decision-making in agriculture, environmental monitoring, and scientific research initiatives. Moreover, the ease of printing RFID and ultra-wideband antennas [109], a well-researched aspect of RFID technology [110], further enhances the feasibility and scalability of implementing RFID-based sensor systems for soil monitoring [111].

Despite significant advancements in energy harvesting and power management technologies, several challenges remain to be addressed before widespread adoption in soil sensing applications can occur. These challenges include improving the efficiency and reliability of energy harvesting modules, enhancing the energy storage capacity of printed batteries and supercapacitors, and optimizing power management algorithms for dynamic energy harvesting conditions. For future research efforts, overcoming these challenges is a critical step toward transforming printed electronic sensor systems into practical applications for soil monitoring and precision agriculture.

### 3.3. Energy Integration Platform

Data transmission technologies play a crucial role in facilitating real-time communication between sensor nodes and data acquisition systems. Bluetooth Low Energy (BLE) stands out as a common wireless communication technique for simple voltage data transmission from electrochemical sensors [112,113]. Wireless sensor networks, using low-power wide area networks (LPWANs) and Zigbee [114], are designed to support low data rate transmission over large areas, fitting the demand for soil sensing. LoRaWAN, a subclass of LPWAN, has been demonstrated successfully for smart agriculture [115]. Jawad et al. present an integrated solution by employing Zigbee to form a wireless sensor network [116]. Figure 8 illustrates the integration of IoT technology within farm fields, supported by a solar cell battery charger. The image depicts an agriculture sensor node featuring an array of sensors powered by solar energy, sink and actuator nodes responsible for efficiently collecting data and executing tasks, and a gateway node facilitating seamless connectivity to cloud computing resources for comprehensive data analysis and management. These technologies offer low-power, long-range communication capabilities, making them suitable for remote soil monitoring applications.

Flexible hybrid electronics (FHE) and Internet of Things (IoT) technologies are incorporated in the integration of energy harvesting capabilities with printed electrochemical sensors for soil monitoring applications. FHE enables the fabrication of flexible and conformable electronic devices, allowing for seamless integration with soil sensor systems and energy harvesting components [117]. IoT platforms facilitate connectivity and data exchange between distributed sensor nodes, enabling scalable and interoperable soil monitoring networks [118].

Combining FHE and IoT technologies enables the development of smart soil monitoring systems capable of autonomously harvesting energy, acquiring data, and transmitting information to central databases or cloud-based platforms. These systems offer real-time insights into soil conditions, enabling proactive decision-making and optimization in agricultural practices, environmental remediation, and scientific research.

## 4. Printed Soil Sensors: Emerging Trends and Outlook

As the global population continues to grow, the demand for food and agricultural products escalates. The availability of farming land, however, faces constraints due to competition from various human activities, including urbanization, desertification, land degradation, and environmental pollution [119]. From individual land farmers to industrialized farm managers, the challenge lies in meeting market demands while preserving the natural environment [120]. Sustainable production goals can be achieved by embracing novel management practices and leveraging new technological advancements for monitoring soil conditions.

Conventional soil sensing techniques often rely on bulky and costly equipment, such as electromagnetic induction sensors, or involve time-consuming soil sampling followed by laboratory soil analysis. In contrast, the application of printed electronics in soil sensors offers advantages over traditional methods. Printed electronics enable the development of compact and lightweight sensors that are easy-to-use in installation and operation [121]. The lightweight nature of printed devices makes them well-suited for portable proximal soil sensing applications [122]. Moreover, the inherent flexibility of printable sensors ensures ease of integration onto curved surfaces and seamless conformability to diverse soil interfaces [123]. Printed electronics offer practical economic benefits by enabling cost-effective production methods, resulting in the creation of affordable sensors suitable for widespread deployment across fields. Additionally, these sensors often utilize organic and biodegradable materials, aligning with the growing emphasis on eco-friendly solutions in agriculture [124]. The production process of printed electronics can be highly efficient, leading to reduced material waste and energy consumption, thereby contributing to both economic savings and environmental stewardship.

The manufacturing prospects of printed electronics for soil sensors are promising due to their inherent advantages in scalability and customization. Unlike traditional manufacturing methods, which often involve complex and expensive fabrication processes, printed electronics can be produced using high-throughput printing techniques on flexible substrates, enabling rapid and cost-effective large-scale production [121]. This scalability is crucial for meeting the growing demand for sensor networks in precision agriculture, where spatial coverage and resolution are essential for accurate monitoring and decision-making [28]. Moreover, the capability to customize sensor design and functionalities through printing processes enables tailoring to specific soil conditions and monitoring targets, enhancing the efficacy and versatility of soil sensing technologies.

Recently, there has been a rise in studies developing printed sensors capable of monitoring soil moisture [125,126], pollutants [40,59], and nutrient levels [26,127] with high sensitivity. Parts of the sensors offer real-time data acquisition and wireless communication capabilities, enabling remote monitoring and control of agricultural systems. Furthermore, the integration of printed sensors with data analytics platforms and machine learning algorithms enhances the predictive capabilities of soil sensing systems, enabling proactive management practices that optimize fertilizer utilization and crop productivity. The cumulative efforts of researchers highlight the potential of printed electronics to revolutionize soil sensing technologies and contribute to sustainable agriculture practices in the future.

### 4.1. Overview of the Soil Sensor Market

Soil sensors enable users to obtain accurate measurements of multiple soil parameters, thereby boosting crop growth, improving the efficiency of fertilizer usage, and promoting environmental stewardship. The increasing adoption of precision agriculture and smart farming among farmers is driving demand for soil sensors, thereby triggering growth in the soil sensor market over the next few years [28]. Among the various types of soil sensors, moisture sensors are considered mature applications and are a major driver of market growth. The global soil moisture sensors market was valued at USD 167.4 million in 2021 and is expected to reach USD 544.0 million by 2030 [128].

With the growing demand for crop products, the price of fertilizers is increasing due to their increased usage to ensure sufficient production [129]. Soil nutrient sensors are expected to be critical tools for monitoring nutrient levels to maintain fertilizer usage efficiency and consider profitability [130]. Meticulous Research projects that the global precision agriculture market will reach USD 27.81 billion by 2031 [131].

Soil measurements often require deploying a large number of sensors at different sites and depths to provide spatially distributed data across fields [132]. Additionally, there is a need for the development of a communication network to transmit real-time monitoring data from sensors to users. Printed electronics offer benefits such as the possibility of integration with communication electronic units [133,134]. Figure 9 illustrates a SWOT analysis (Strengths, Weaknesses, Opportunities, and Threats) of printed electrochemical sensors for soil sensing applications. Printed electrochemical sensors combine material and mechanical characteristics from printing electronics with strong sensing capabilities from electrochemical sensing, showing great strengths and opportunities in soil sensing.

### 4.2. Emerging Demands in Agricultural Applications

In the past, farmers obtained soil information through soil sampling and laboratory analysis. This process is time-consuming and is limited to a certain number of sampling sites considering parameters such as nutrient levels, moisture content, and pH are varied due to spatial heterogeneity and farming practices. Access to accurate data from various field sites is crucial for optimal decision-making. In situ sensors can continuously provide precise data across fields, enabling farmers to make informed decisions on fertilizer applications and crop yield optimization. However, Fan et al. underscored that despite over 4000 publications on soil monitoring sensors between 2000 and 2020, only 558 enabled real-time monitoring, with a mere 67 focusing on continuous in situ soil measurement [5]. Another obstacle hindering the adoption of new technologies is the high cost of sensors. Printing technology has emerged as a promising solution to reduce the fabrication costs of sensors. The rising demand from farmers underscores the importance of developing sensors that are economically feasible and capable of being deployed into fields to harvest spatiotemporal soil information, ultimately implementing effective management strategies in field production.

The real-time monitoring of nutrients has the potential to directly impact the economics of farming. The aforementioned, N, P, and K are key nutrients that are strongly related to plant growth and crop yield. Researchers have worked on the development of a multifunctional sensor platform that allows them to sense these macronutrients at the same time [24,135]. Several publications have utilized technologies depending on nanotechnology [102,103], MEMS [136], and printed technology [137]. Madhumathi et al. demonstrated an IoT system for the connectivity of multiple sensors [138]. Dattatreya et al. argue that when data is collected in the cloud, machine learning and deep learning can be applied for calibration and prediction modeling [135]. Ultimately, utilizing real time monitoring to enable more precise nutrient management will allow for higher crop yields while simultaneously reducing environmental damage.

From an environmental conservation perspective, embedded soil sensors could potentially impact the soil environment. The development of environmentally friendly ink has become a prominent area of research. Sui et al. introduced a biodegradable moisture sensor featuring printed Zn interdigitated electrodes (IDE) on a poly(3-hydroxybutyrate-co-3-hydroxyvalerate) (PHBV) film [88]. This sensor can operate in the field for up to 30 days before initiating decomposition in the soil environment. Polymer materials have been extensively studied in printing technology, including conventional 3D printing materials such aspolyethylene (PE), polyimide (PI), thermoplastic, and polylactic acid (PLA) [124,139]. Additionally, materials suitable for substrates in sensor devices, such as poly(methyl methacrylate) (PMMA), polyetherimide (PEI), polycarbonate (PC), and poly(ether ether ketone) (PEEK), have demonstrated biodegradability and responsiveness to specific stimuli [140]. Over the past decade, 4D printing has emerged as a novel technology, offering materials capable of responding to environmental stimuli. With advancements in both 3D and 4D printing techniques, there is potential to integrate them with state-of-the-art methods for fabricating fully printed biodegradable sensor devices.

### 4.3. Remaining Challenges for Printed Soil Sensors

Despite the promising potential demonstrated in the aforementioned publications, printed soil sensors still encounter several critical challenges awaiting solutions, as shown in Figure 9. Ensuring the durability and longevity of printed sensors in harsh soil environments remains a priority. Printed biodegradable sensors typically begin to degrade after approximately 30 days, posing a significant obstacle to the long-term monitoring goal. Therefore, there is a pressing need to develop strategies for continuous monitoring throughout the entire plant season to ensure the sensors remain functional and reliable.

Printed IDE moisture sensors are at the forefront of soil sensor technology, capable of being employed across various environments while providing accurate moisture content data. However, other sensors have not demonstrated the same level of universality and precision. Most sensors typically require a one-point or two-point calibration process to ensure accuracy. Sensor calibration presents another significant obstacle, as it is often a time- and labor-intensive process. Given the diverse types of soil textures, calibration for each site becomes necessary to maintain accuracy. While sensors may maintain their performance over months, regular calibration is necessary to maintain high levels of precise measurements. Simplifying and cost-effectively streamlining the calibration procedure is imperative for widespread adoption.

Effectively translating sensor data into comprehensive insights for farmers is essential. This involves the development of user-friendly interfaces, algorithms, and decision support tools that enable farmers to make informed decisions based on sensor data. However, there is currently a gap in understanding soil nutrient kinetics, hindering the ability to provide comprehensive insights. Further research is necessary to develop modules to address this knowledge gap. Addressing these issues relies heavily on research efforts to create a user-friendly soil monitoring system that meets the needs of farmers and agricultural professionals.

## 5. Conclusions

Printed sensors stand out as a promising solution to meet the rising demand for precision agriculture. Potentiometric sensors are successfully utilized for pH and ion concentration measurements. However, they need to be calibrated regularly. Voltametric sensors are highly sensitive and selective, capable of sensing multiple targets, but they require complex data analysis. Amperometric sensors have high sensitivity and selectivity but may face electrode fouling over time as a result of enzyme activity. Impedimetric sensors enable label-free detection, albeit with lower sensitivity. Each electrochemical sensing method offers its own set of advantages and limitations in detecting analytes. Despite their differences, these sensors exhibit versatile capabilities in capturing soil information, enabling real-time continuous measurements. The availability of this data is crucial for decision-makers in multiple fields, including crop yield enhancement, precision agriculture, and environmental stewardship.

Moreover, self-powered solutions and seamless communication with users have become imperative requirements. The integration potential of printed sensors with other electronic components, namely through FHE, presents an exciting avenue for future soil monitoring efforts. Printed electronics enable the ability to include components with data communication functions and energy harvesting capabilities. By combining printed sensors with hybrid electronics, a practical approach for soil monitoring arises, offering both scalability and economic feasibility for more accurately managing agricultural practices.

## Figures and Tables

**Figure 1 micromachines-15-00625-f001:**
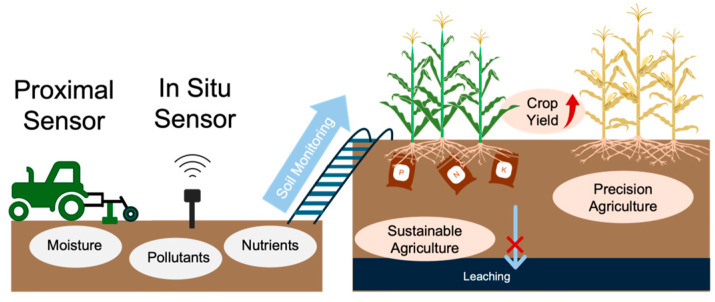
Soil monitoring provides benefits for more effective agricultural management.

**Figure 2 micromachines-15-00625-f002:**
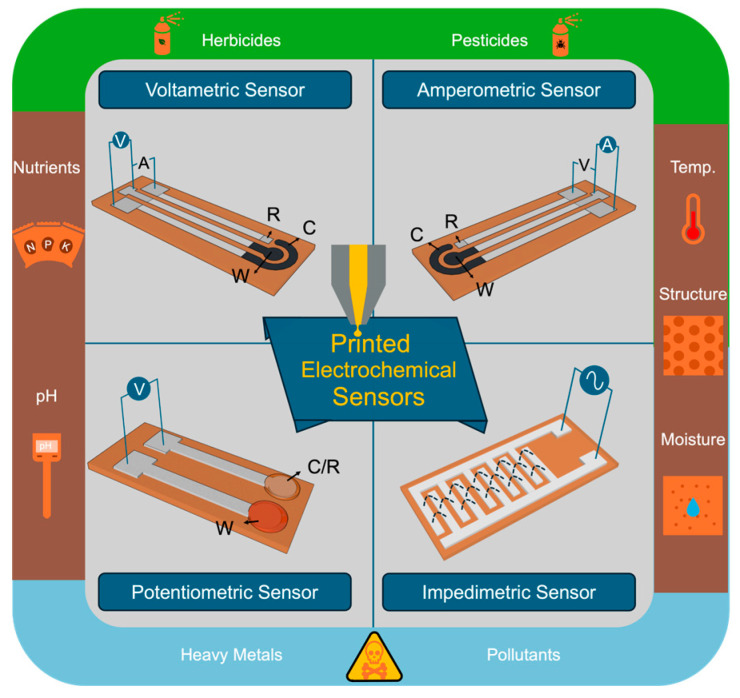
Printed technologies enable electrochemical sensors for soil monitoring, showcasing their versatility and potential in soil sensing applications. W denotes for working electrode, R stands for reference electrode, and C represents counter electrode.

**Figure 3 micromachines-15-00625-f003:**
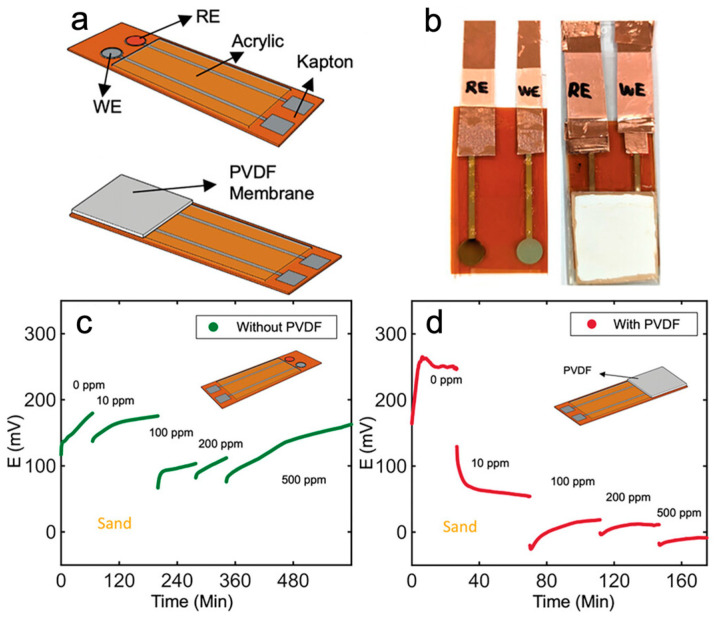
Printed potentiometric sensor for soil nitrate detection. (**a**) Schematic and (**b**) photograph of an inkjet-printed potentiometric sensor featuring Ag/AgCl working (WE) and a reference electrode (RE), with a PVDF membrane coated on the sensing area. (**c**,**d**) Open-circuit potentiometry measurements in sandy soil with nitrate concentrations ranging from 0 ppm to 500 ppm (**c**) without PVDF and (**d**) with PVDF. The sensor with PVDF provides steady potential readings within a shorter time compared to the sensor without PVDF. (**a**–**d**) Reproduced with permission. Copyright 2024, Wiley [26].

**Figure 4 micromachines-15-00625-f004:**
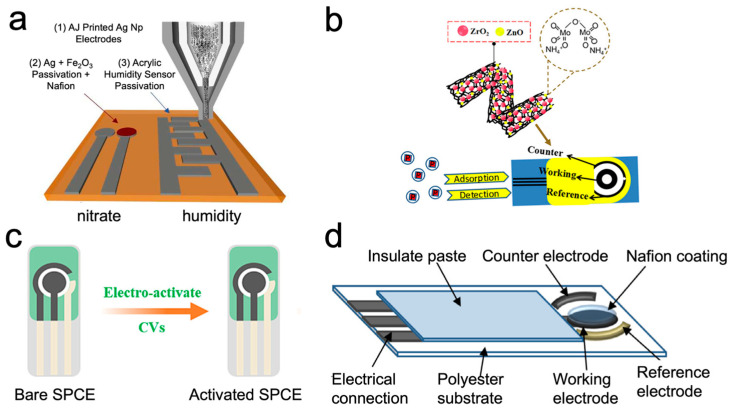
Printed sensors for ion detection in soil using voltammetry. (**a**) Dual aerosol jet-printed humidity and nitrate sensors. (**b**) Phosphate sensors constructed using zirconium dioxide/zinc oxide/multiple-wall carbon nanotubes/ammonium molybdate tetrahydrate/screen printed electrodes (ZrO_2_/ZnO/MWCNTs/AMT/SPE). (**c**) Screen-printed electrodes modified with silica isoporous membranes (SIMs) using electrochemically assisted self-assembly (EASA) for the simultaneous detection of heavy metal ions. (**d**) Screen-printed electrodes using carbon ink (working, counter) and Ag/AgCl (reference) modified with Nafion and Bismuth for the simultaneous detection of Cd^2+^ and Pb^2+^ ions. (**a**) Reproduced with permission. Copyright 2021, IEEE [48]. (**b**) Reproduced with permission. Copyright 2021, ACS [51]. (**c**) Reproduced with permission. Copyright 2023, Elsevier [59]. (**d**) Reproduced with permission. Copyright 2016, Elsevier [60].

**Figure 5 micromachines-15-00625-f005:**
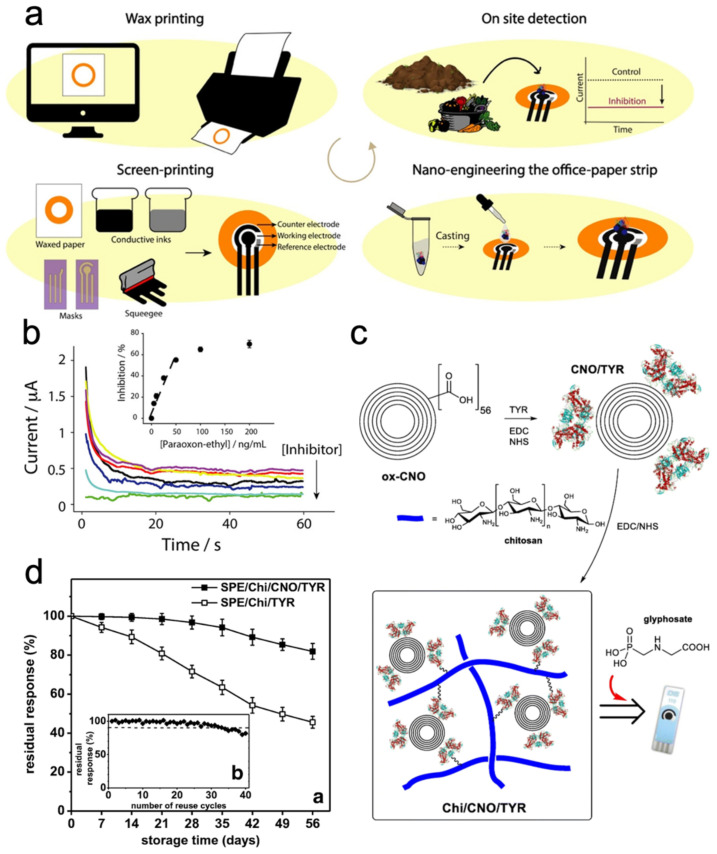
Printed soil amperometry sensors. (**a**) The illustration depicts the fabrication process, sample test procedure, and signal response of an office paper-based screen-printed enzyme-inhibition amperometric Sensor designed for detecting Organophosphate pesticides; (**b**) The recorded chronoamperometric data ranges from 0 to 200 ng/mL of paraoxon-ethyl. The inset displays the calibration curve using the same concentration of the inhibitor; (**c**) the manufacturing process of the Chi/CNO/TYR modified screen-printed sensor utilized for detecting glyphosate herbicide; (**d**) residual activity graphs demonstrate enhanced stability and repeatability of the sensor with the addition of the nanomaterial CNO. (**a**,**b**) Reproduced with permission. Copyright 2021 [53], American Chemical Society. (**c**,**d**) Reproduced with permission. Copyright 2019 [77], Springer.

**Figure 6 micromachines-15-00625-f006:**
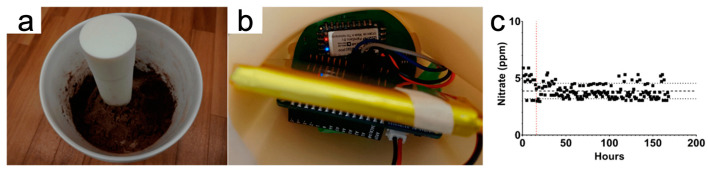
In situ electrochemical soil nitrate sensor sensed by the EIS method. (**a**,**b**) Sensor and processing electronic hardware set up in sandy loam soil; (**c**) In situ EIS measurement over 7 days in soil. (**a**–**c**) Reproduced with permission.

**Figure 7 micromachines-15-00625-f007:**
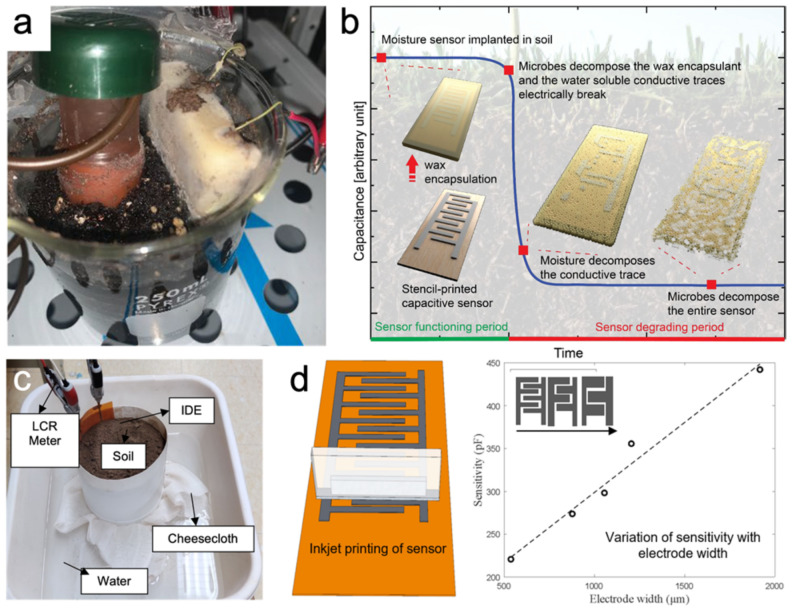
Soil moisture sensors enabled by the IDE pattern. (**a**,**c**) Set up for real-time soil moisture measurement; (**b**) Illustration of the degradation mechanism of the moisture sensor over time; (**d**) The sensitivity of the IDE sensor is proportional to the width of the interdigitated finger. (**a**,**b**) Reprinted with permission from [88]. Copyright 2021, American Chemical Society. (**c**,**d**) Reproduced with permission. Copyright 2022 [89], IEEE.

**Figure 8 micromachines-15-00625-f008:**
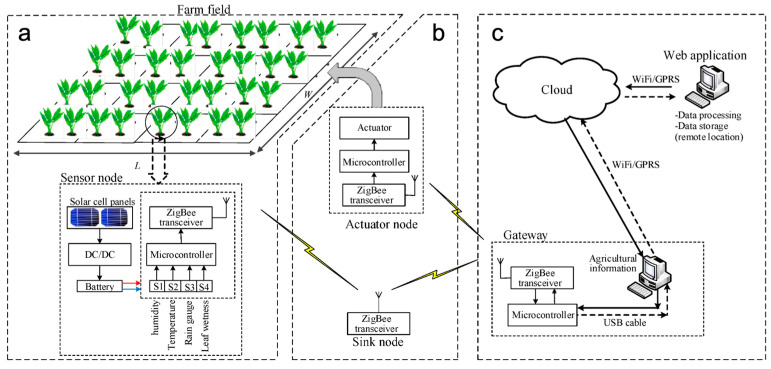
Integration of IoT in farm fields facilitated by a solar cell battery charger: (**a**) Illustration of an agriculture sensor node equipped with various sensors powered by solar energy; (**b**) Sink and actuator nodes orchestrating data collection and task execution; and (**c**) Gateway nodes enabling seamless connectivity to cloud computing for extensive data analysis and management [116]. (**a**–**c**) Reproduced with permission.

**Figure 9 micromachines-15-00625-f009:**
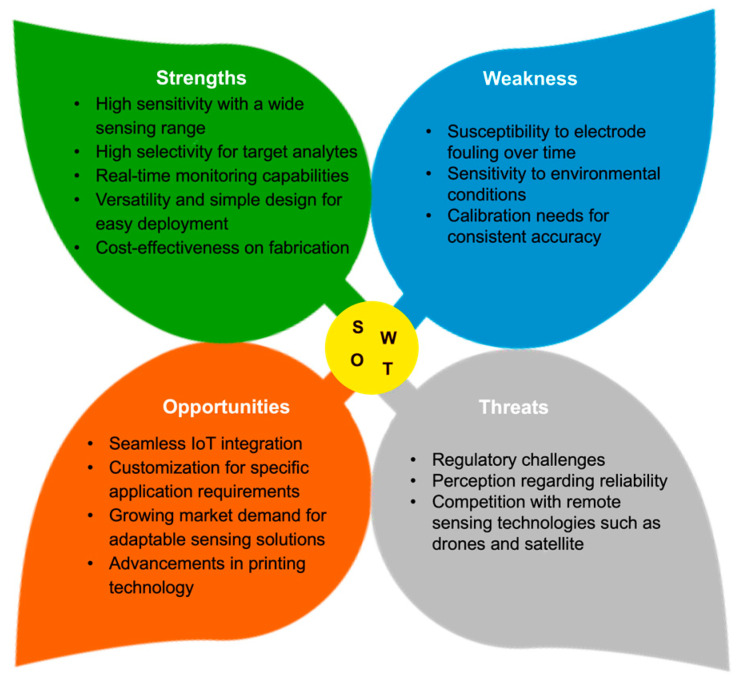
A detailed SWOT analysis of printed electrochemical sensors for soil sensing.

**Table 1 micromachines-15-00625-t001:** Potentiometric sensors in soil monitoring applications.

Printing Techniques/Fabrication Approach	Working Electrode	Functionalization Methods	Target	Soil Test (Moisture)	Sensitivity (mV/dec)	Linear Detection Range (ppm)	LOD (ppm)	Ref.
Transfer/Laser	Graphene	Nonactin/TCPB/DOS/PVC	${\mathrm{NH}}_{4}^{+}$	Soil Slurry (1:10 *w*/*w*)	51.7	0.18 to 1800	0.5	[38]
		Catonium MTPB/ nitrocellulose/IPA /NPOE/TDMAN/PVC/THF	${\mathrm{NO}}_{3}^{–}$		−54.8	0.62 to 6200	1.3	
E-beam Deposition Fluid Dispensing	Au POT-MoS_2_	Catonium MTPB/ nitrocellulose/IPA /NPOE/TDMAN/PVC/THF	${\mathrm{NO}}_{3}^{–}-N$	Soil Slurry	−64	1 to 1500	1.3	[39]
SPE	Ag	Ag/AgCl	${\mathrm{Cl}}^{–}$	Sterile sandy clay loam soil	−48.6	3.55 to 7091	-	[35]
SPE	Carbon	PVC/NPOE/MIP+PANI	2,4-D	Orchard soil	22.08	2.21 to 22.10	0.1	[40]
IJP	Gold	Nitrate Ionophore VI/PVC/DBP/TBAC/THF	${\mathrm{NO}}_{3}^{–}$	Peat soil (2.8:10 *w*/*w*)	−47	6.2 to 6200	3.1	[41]
IJP	Ag	TOA-bromide/PVC/plasticizer/THF	${\mathrm{NO}}_{3}^{–}$	Soil (6:10 *w*/*w*)	−50~−52	62 to 6200	-	[25]
IJP	Ag/AgCl	NPOE/PVC/TDAN/THF	${\mathrm{NO}}_{3}^{–}$	Sandy soil (4:10 *w*/*w*)	−56.29	10 to 200	8.8	[26]
				Silt loam soil (4:10 *w*/*w*)	−46.56			

Note: SPE stands for screen-printed electrode, IJP represents inkjet printing technique, 2,4-D denotes 2,4-Dichlorophenoxyacetic acid, LOD indicates the limit of detection.

**Table 2 micromachines-15-00625-t002:** Recent publications on printed voltametric sensors for soil nutrient monitoring and soil pollutant detection.

Target Ions	Printing Technique	Functionalization Method	Measurement Method	LOD (ppm)	Linear Detection Range (unit)	RSD	Ref.
${\mathrm{NO}}_{3}^{–}$	SPE	CdS-NRs	CV	0.14	3.10–310	1.6%	[23]
	SPE	BSA/NiR/CNT-α-Fe_2_O_3_	CV	0.09	0.05–500	0.43%	[47]
	AJP	Ag-Fe_2_O_3_/Nafion	CV	-	1–400	-	[48]
$N^{–}$	SPE	MWCNT-CS	CV/Amperometry	0.11	0.74–78.2	5.70%	[49]
	3D printing	PLA filament	CV/SWV	0.09	1.15–69	<7%	[50]
${\mathrm{PO}}_{4}^{3–}$	SPE	ZrO_2_/ZnO/MWCNTs/AMT	CV	0.0019	0.0035–0.1	5.80%	[51]
	SPE	CBNPs	CV	0.4	0.47–9.5	5.30%	[52]
	SPE	Prussian blue, carbon black, and butyrylcholinesterase	CV/Chronoamperometry	0.0013	0–0.025	-	[53]
$K^{+}$	SPE	Nafion/4-aminobenzo-18-crown-6 ether	CV/DPV	-	1–500	2.45%	[54]
	SPE	PEDOT:PSS/KNO_3_ gel	CV	-	0.39–390	-	[24]
Pb^2+^	SPE	Au@Py	CV	0.0006	0.0005–0.025	4.78%	[55]
Pb^2+^ Cd^2+^	SPE	2% Bi_2_O_3_/Graphite carbon ink	CV/Chronopotentiometry	0.008 0.016	0.002–0.3	5.60% 9.10%	[56]
Cu^2+^	SPE	Hg	SWASV	0.0055	-	6%	[57]
Hg^2+^	SPE	CBNP-AuNPs	SWASV	0.003	0–0.1	<10%	[58]
Cd^2+^ Pb^2+^ Cu^2+^ Hg^2+^	SPE	SIM	SWASV	0.001 0.0002 0.001 0.0002	0.02–2.28 0.002–2.07 0.01–1.27 0.002–2.01	<3%	[59]
Pb^2+^ Cd^2+^	SPE	Nafion/Bi film	DPASV	0.0016 0.0025	0–0.08	3.5% 3.8%	[60]

Note: SPE refers to screen printed electrode, AJP denotes aerosol jet printing, CV refers to cyclic voltammetry, SWASV stands for Square Wave Anodic Stripping Voltammetry, DPASV stands for Differential Pulse Anodic Stripping Voltammetry, LOD stands for limit of detection, and RSD represents relative standard deviation.

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
