# Peer review of "Underground Ink: Printed Electronics Enabling Electrochemical Sensing in Soil"

_micromachines, 2024, doi:10.3390/mi15050625_

Round 1
Reviewer 1 Report
Comments and Suggestions for Authors
This review provides a comprehensive overview of recent work related to printed electrochemical soil sensors, and provides insight into future research directions which can enable widespread adoption of precision agriculture technologies. It is promising. However, (1) the references need to be check carefully, such as Analytical Methods in line 850 uses the full name of the journal, but Anal Chim Acta in 86 is simplified name. (2) LOD needs to be provided in the table for detection. (3) The authors had better give more comparsion and discussion for the different methods.
Comments on the Quality of English LanguageEnglishi is good, and it only needs to be checked.
Reviewer 2 Report
Comments and Suggestions for Authors
Manuscript title: Underground Ink: Printed Electronics Enabling 2 Electrochemical Sensing in Soil
Manuscript ID: micromachines-2975984
The authors thoroughly review the recent literature in the area of flexible printed electronics used for soil sensing using electrochemical techniques. The proposed topic is pretty interesting and realistic for agricultural applications for sensing and testing soil nutrients with conductive printed electrodes that can be the environmentally friendly approach for university point-of-source applications. This review also throws limelight on the future directions that explore the advanced technology in agricultural sciences. The manuscript is well-written and well-organized discussing comprehensively with eye-catching figures. This also falls well within the scope of the journal.
However, the manuscript can be further improved and enhanced by adding a few more insightful information. Below are the minor comments:
1) Please remove the incomplete ‘@wisc.edu’ associated with the affiliation.
2) Abstract should be strengthened by adding a few key points to the proposed review.
3) Significance of flexible printed electronics in the field of agriculture sciences.
4) Section 2 should be elaborated with more recent literature discussing the several fabrication techniques involved in the development of printed EC sensors.
5) Authors should consider adding these relevant references to the manuscript: doi.org/10.1016/j.colsurfb.2021.112056; doi.org/10.3390/s23073692; 10.1039/D4AY00293H; 10.1109/TAFE.2024.3351953; doi.org/10.1016/j.sna.2021.112896; doi.org/10.1002/elan.202200202;
6) What is the impact of material selection? Pros and cons?
7) Page 17, line 661. ‘MEMs’ should be ‘MEMS’
8) Add recent references covering the last 2-3 years in the proposed area.
9) Please discuss different detection methods that are applicable in printed flexible sensors.
10) Please add the SWOT analysis of printed EC sensors by providing the schematic diagram/ figure.
11) What are the challenges and limitations? Why there is a hindrance in commercializing the product?
12) Please sensitivity, specificity, LOD, and reproducibility.
13) Overall, the language needs to be polished and in-flow.
Comments on the Quality of English LanguageAttached
